# Mesenchymal Stem Cell-Derived Small Extracellular Vesicles Protect Cardiomyocytes from Doxorubicin-Induced Cardiomyopathy by Upregulating Survivin Expression via the miR-199a-3p-Akt-Sp1/p53 Signaling Pathway

**DOI:** 10.3390/ijms22137102

**Published:** 2021-07-01

**Authors:** Ji Yoon Lee, Jihwa Chung, Yeongju Byun, Kyoung Hwa Kim, Shung Hyun An, Kihwan Kwon

**Affiliations:** 1Medical Research Institute, School of Medicine, Ewha Womans University, Seoul 07985, Korea; ljy0404@gmail.com (J.Y.L.); byunyj@ewhain.net (Y.B.); 2Exollence Biotechnology Co., Ltd., Seoul 07985, Korea; jhchung@exollence.com (J.C.); khkim@exollence.com (K.H.K.); shan@exollence.com (S.H.A.); 3Department of Internal Medicine, Cardiology Division, School of Medicine, Ewha Womans University, Seoul 07985, Korea

**Keywords:** MSC-sEVs, doxorubicin-induced cardiomyopathy, survivin, miRNA, apoptosis

## Abstract

Cardiotoxicity is associated with the long-term clinical application of doxorubicin (DOX) in cancer patients. Mesenchymal stem cell-derived small extracellular vesicles (MSC-sEVs) including exosomes have been suggested for the treatment of various diseases, including ischemic diseases. However, the effects and functional mechanism of MSC-sEVs in DOX-induced cardiomyopathy have not been clarified. Here, MSC-sEVs were isolated from murine embryonic mesenchymal progenitor cell (C3H/10T1/2) culture media, using ultrafiltration. H9c2 cardiac myoblast cells were pretreated with MSC-sEVs and then exposed to DOX. For in vivo studies, male C57BL/6 mice were administered MSC-sEVs intravenously, prior to a single dose of DOX (15 mg/kg, intraperitoneal). The mice were sacrificed 14 days after DOX treatment. The results showed that MSC-sEVs protected cardiomyocytes from DOX-induced cell death. H9c2 cells treated with DOX showed downregulation of both phosphorylated Akt and survivin, whereas the treatment of MSC-sEVs recovered expression, indicating their anti-apoptotic effects. Three microRNAs (miRNAs) (miR 199a-3p, miR 424-5p, and miR 21-5p) in MSC-sEVs regulated the Akt-Sp1/p53 signaling pathway in cardiomyocytes. Among them, miR 199a-3p was involved in regulating survivin expression, which correlated with the anti-apoptotic effects of MSC-sEVs. In in vivo studies, the echocardiographic results showed that the group treated with MSC-sEVs recovered from DOX-induced cardiomyopathy, showing improvement of both the left ventricle fraction and ejection fraction. MSC-sEVs treatment also increased both survivin and B-cell lymphoma 2 expression in heart tissue compared to the DOX group. Our results demonstrate that MSC-sEVs have protective effects against DOX-induced cardiomyopathy by upregulating survivin expression, which is mediated by the regulation of Akt activation by miRNAs in MSC-sEVs. Thus, MSC-sEVs may be a novel therapy for the prevention of DOX-induced cardiomyopathy.

## 1. Introduction

Doxorubicin (DOX) is a type of anthracycline that is widely used to treat various cancers [1,2]. While it is a highly effective anti-cancer agent, cardiotoxicity is a well-known side effect associated with its long-term clinical use. Treatment with DOX induces cardiomyocyte apoptosis, followed by increasing oxidative stress [1,3]. Likewise, a cumulative dose of DOX in an in vivo mouse model was shown to induce cardiotoxicity, with cardiac symptoms similar to those of dilated cardiomyopathy; the cardiac chambers dilated and the ventricular ejection fraction (EF) and contractile function were significantly reduced [4,5,6]. Thus, it is necessary to identify effective methods to prevent the cardiotoxicity of DOX in cancer chemotherapy.

Survivin is a member of the inhibitor of apoptosis protein (IAP) family, which regulates cellular apoptosis and tumor progression in various cell types [7,8,9]. As the administration of DOX results in cardiomyocyte apoptosis, survivin is considered a novel therapeutic target in patients with DOX-induced cardiomyopathy. Several studies confirmed that delivery of recombinant survivin to H9c2 cardiomyocytes induces an anti-apoptotic effect against DOX-induced apoptosis [10,11]. However, providing exogenous survivin for therapeutic purposes over the long term is not practical due to its short half-life [12]. Therefore, upregulation of endogenous survivin may be a more effective therapy against DOX-induced cardiomyopathy.

Exosomes are small extracellular vesicles (sEVs), with a diameter between 30 and 100 nm, that are secreted from various cell types and harbor complex contents including proteins, lipids, growth factors, and microRNAs (miRNAs) [13,14]. In particular, mesenchymal stem cell-derived sEVs (MSC-sEVs) have therapeutic effects on cardiac injuries, due to their regeneration and remodeling abilities [15]. Indeed, previous studies have demonstrated that MSC-sEVs have protective effects on myocardial infarction/reperfusion injury by inducing angiogenesis [16]. Based on these results, we hypothesized that they would also have protective effects on cellular apoptosis by doxorubicin-induced cardiotoxicity.

MiRNAs are a class of small non-coding RNAs of approximately 20 nucleotides in length. They act as negative regulators of gene expression by targeting specific mRNA sequences, resulting in mRNA degradation and translational repression [17,18], and modulate complex pathophysiology by regulating various protein-encoding genes associated with cellular functions [19]. In particular, miRNAs are involved in the regulation of cardiac function, including apoptosis, electrical signals, and heart development, and are considered potential therapeutic targets for cardiac diseases [20,21].

Therefore, recent studies have focused on effective methods for the delivery of miRNAs derived from MSCs to cardiac cells rather than the implantation of MSCs directly into cardiac tissues, due to their short lifetime and limited efficacy. This study investigated the potential cardioprotective effects of MSC-sEVs against DOX-induced cardiotoxicity and the mechanism underlying survivin induction in both in vitro and in vivo models.

## 2. Results

### 2.1. Characterization of MSC-sEVs

To confirm the purification of MSC-sEVs, isolated MSC-sEVs were validated by their size distribution, number, morphology and exosome markers (Figure A1). The NTA results showed that isolated MSC-sEVs had a peak distribution < 200 nm in diameter, consistent with the representative size of pure sEVs. The hydrodynamic diameter of sEVs measured using NTA was approximately 135 nm (Figure A1A). Next, the morphology of MSC-sEVs was observed via AFM. The images obtained by AFM showed the round-shaped structure of exosomes (Figure A1B). In addition, MSC-derived exosome markers, including the cluster of differentiation 9 (CD9), CD63, CD81, and tumor susceptibility gene 101 (TSG101), were detected by Western blotting (Figure A1C). The isolated sEVs from C3H10T1/2 were indeed exosomes, showing representative characteristics.

### 2.2. MSC-sEVs Protect Cardiomyocytes from DOX-Induced Cell Death

To determine the effects of MSC-sEVs on the DOX-induced apoptosis of cardiomyocytes, both cell viability and the level of apoptosis were measured. As is consistent with a previous study [22,23], treatment with DOX significantly reduced H9c2 cell viability. However, treatment with MSC-sEVs prior to DOX treatment restored the suppressed cell viability (Figure 1A). Similarly, the TUNEL assay showed that DOX treatment induced cellular apoptosis, whereas pretreatment with MSC-sEVs alleviated it (Figure 1B). In addition, we measured the protein level of Bcl-2, a representative anti-apoptotic protein. As shown in Figure 1C, Bcl-2 protein levels were lower in DOX-treated cells than in control cells, whereas stimulation with MSC-sEVs prior to DOX treatment enhanced the suppressed Bcl-2 expression by DOX. These results indicate that MSC-sEVs protect cardiomyocytes against DOX-induced cellular apoptosis.

### 2.3. MSC-sEVs Inhibit DOX-Induced Downregulation of Survivin via the Akt Pathway in Cardiomyocytes

PI3K/Akt signaling is involved in survivin expression in cardiomyocytes [7,24]. To determine whether Akt phosphorylation is regulated by MSC-sEVs stimulation, H9c2 cells were treated with MSC-sEVs and incubated for the indicated time periods. As shown in Figure 2A, MSC-sEVs stimulation upregulated p-Akt levels, whereas total Akt expression remained constant; the highest p-Akt/total Akt levels were observed 1 h after MSC-sEVs treatment. Protein levels of survivin also increased and peaked in cells 24 h after exposure to MSC-sEVs. As is consistent with previous studies [10,22,25], the protein levels of survivin and Bcl-2 were significantly downregulated in H9c2 cells treated with DOX for 24 h (Figure A2). We hypothesized that MSC-sEVs increased the expression of survivin, which had been downregulated by DOX via the Akt pathway. H9c2 cells were treated with MSC-sEVs 24 h prior to DOX treatment for the indicated time points, to determine if MSC-sEVs could recover survivin levels by phosphorylating Akt under the DOX-treated condition. DOX treatment suppressed both p-Akt and survivin levels in cardiomyocytes, whereas exposing MSC-sEVs to cells prior to DOX treatment recovered these levels (Figure 2B,C). Next, to determine whether MSC-sEVs upregulate survivin through Akt activation, we inhibited Akt phosphorylation using a PI3K inhibitor, LY294002. We confirmed that 50 μM LY294002 significantly suppressed the increased level of p-Akt by MSC-sEVs treatment in H9c2 cells (Figure A3A). As shown in Figure A3B, treatment with a PI3K inhibitor at this concentration significantly inhibited the elevated expression of survivin induced by MSC-sEVs, suggesting that Akt activation by MSC-sEVs induces survivin expression in cardiomyocytes. Furthermore, the inhibition of Akt using LY294002 reduced survivin levels, which were recovered by MSC-sEVs in DOX-treated cells (Figure 2D). These results suggest that MSC-sEVs restore the survivin expression reduced by DOX via Akt activation in cardiomyocytes.

### 2.4. Knockdown of Survivin Alleviates the Anti-Apoptotic Effects of MSC-sEVs in DOX-Induced Cell Death

To clarify whether survivin induced by MSC-sEVs contributes to the anti-apoptotic effect on DOX-induced cell death in cardiomyocytes, survivin levels were suppressed by siRNA transfection in cardiomyocytes, and the effects of MSC-sEVs on DOX-treated cardiomyocytes were observed. Survivin was effectively suppressed by transfection of target siRNA (Figure A4). After siRNA transfection, H9c2 cells were exposed to MSC-sEVs for 24 h, followed by DOX treatment for another 24 h. As shown in Figure 3A, the level of recovered anti-apoptotic Bcl-2 by MSC-sEVs in DOX-treated cells decreased following the knockdown of survivin. Cell viability and apoptosis level were measured using the MTT and TUNEL assays, respectively. The results showed that the knockdown of survivin reversed the anti-apoptotic effects of MSC-sEVs in DOX-induced cell death; cell viability decreased, while cellular apoptosis increased (Figure 3B–D). These results suggest that survivin may be the major factor contributing to the anti-apoptotic effects of MSC-sEVs against DOX-induced cell death in cardiomyocytes.

### 2.5. Akt Activation by MSC-sEVs Induces Survivin by Regulating p53 and Sp1 in DOX-Treated Cardiomyocytes

To identify the key transcription factors mediating survivin expression by MSC-sEVs, we tested Sp1 and p53, which regulate survivin expression under various stimuli [24]. The p53 expression is induced by DOX treatment in cardiomyocytes, whereas DOX inhibits Sp1 activation [24,25,26,27]. Based on these studies, we hypothesized that Akt activation by MSC-sEVs regulates both Sp1 and p53 and induces survivin expression. As is consistent with previous studies, the expression levels of both Sp1 and p53 changed in response to DOX treatment in H9c2 cells; Sp1 levels decreased and p53 levels increased following DOX treatment (Figure A5). Furthermore, we investigated whether MSC-sEVs treatment affects the activation of Sp1 and p53 in cardiomyocytes in DOX-treated cells. As shown in Figure 4A, cells exposed to MSC-sEVs reversed the activities of both Sp1 and p53, induced by DOX treatment in H9c2 cells; downregulated Sp1 by DOX significantly increased in cells treated with MSC-sEVs, while upregulated p53, following DOX treatment, decreased. Next, we performed cytosol/nuclear fractionation to determine whether nuclear activation of these two transcription factors is regulated under DOX-treated conditions. The downregulated nuclear translocation of Sp1 by DOX was recovered in cells treated with additional MSC-sEVs. Conversely, the increased nuclear translocation of p53 by DOX decreased after additional MSC-sEVs treatment (Figure 4B). Next, we determined whether the regulation of both Sp1 and p53 by MSC-sEVs is mediated by Akt activation. After treatment (or not) with LY294002 for 2 h, the cells were exposed to MSC-sEVs 24 h prior to DOX treatment. The activities of Sp1 and p53 were reversed in cells treated with additional MSC-sEVs, compared to DOX-treated cells. However, the effects of MSC-sEVs were alleviated in DOX-treated cells when Akt was inhibited by LY294002 (Figure 4C). These results suggest that Akt activation by MSC-sEVs treatment induces Sp1 and inhibits p53 activation, followed by upregulation of survivin expression in DOX-treated cardiomyocytes.

### 2.6. MiRNAs in MSC-sEVs Contribute to the Akt/Survivin Pathway in Cardiomyocytes

MSC exosomes have protective effects on pathogenic conditions via conveying various miRNAs to lesions [15,28]. Because we confirmed that MSC-sEVs alleviated apoptosis induced by DOX in cardiomyocytes, we hypothesized that some types of miRNAs in MSC-sEVs are responsible for the anti-apoptotic effects of MSC-sEVs. Based on previous studies, we selected three miRNAs (miR-199a-3p, miR-424-5p, and miR-21-5p) that are rich in MSC-sEVs and inhibit apoptosis in various cell types, and thus may play key roles in alleviating apoptosis in DOX-treated cardiomyocytes [15,29,30]. To identify whether those three miRNAs are indeed representative of MSC-sEVs compared to sEVs from other types of cells, we compared their levels between immortalized mouse aortic endothelial cells-derived small extracellular vesicles (iMAEC-sEVs) and MSC-sEVs. As shown in Figure 5A, miR-199a-3p levels were higher in MSC-sEVs than in iMAEC-sEVs, whereas the levels of miR-424-5p and miR-21-5p were lower. Next, we measured their levels in MSC-sEVs, which were normalized to the level of cel-miR-39-3p [31]. We found that miR-21-5p was the most abundant, followed by miR-424-5p and miR-199a-3p (Figure 5B). To explore the effects of these miRNAs on cardiomyocytes, we observed changes in the Akt/survivin pathway in H9c2 cells transfected with three individual miRNA mimics. Overexpression of each miRNA mimic significantly upregulated the miRNA level in H9c2 cells (Figure 5C). The levels of p-Akt were higher compared to controls when the three miRNA levels were upregulated individually. Furthermore, overexpression of miR-199a-3p increased the protein level of survivin, whereas overexpression of miR-424-5p and miR-21-5p did not (Figure 5D).

### 2.7. MiR-199a-3p in MSC-sEVs Mediates the Upregulation of Akt/Survivin in DOX-Treated Cardiomyocytes

To confirm whether MSC-sEVs deliver miRNAs such as miR-199a-3p, miR-424-5p, and miR-21-5p to cardiomyocytes in DOX-treated conditions, we measured the levels of these three miRNA in cells treated with MSC-sEVs for 24 h prior to DOX treatment for another 24 h. As shown in Figure 6A, cells exposed to MSC-sEVs before DOX treatment showed higher levels of each miRNA vs. the DOX-treated group. To assess whether these miRNAs regulate the Akt/survivin pathway, we overexpressed each by transfection of miRNA mimics, followed by DOX treatment. As is consistent with previous studies, p-Akt levels decreased in DOX-treated cells compared to control cells, but the overexpression of each individual miRNA mimics restored phosphorylation of Akt downregulated by DOX (Figure 6B). Moreover, overexpression of the miRNAs showed similar effects on transcriptional activity with MSC-sEVs; it induced upregulation of Sp1 and downregulation of p53 in DOX-treated cells (Figure 6C). Furthermore, the overexpression of miR-199a-3p and miR-21-5p reversed the downregulation of survivin by DOX in cardiomyocytes, whereas the overexpression of miR-424-5p showed no significant change in survivin expression (Figure 6B). Based on these results, miR-199a-3p and miR-21-5p were chosen as the major candidate miRNAs in MSC-sEVs regulating survivin level in DOX-treated cardiomyocytes.

To confirm this suspicion, we inhibited each miRNA using an inhibitor. As shown in Figure A6, this effectively blocked the target miRNAs and significantly decreased their levels. This in turn reversed the effects of MSC-sEVs on p-Akt (Figure 7A). In particular, inhibition of miR-199a-3p effectively reversed the effects of MSC-sEVs on Sp1 and p53, resulting in the downregulation of survivin in MSC-sEVs-treated cardiomyocytes. However, there were no significant changes in cells upon the inhibition of miR-21-5p (Figure 7B,C). These results indicate that miR-199a-3p in MSC-sEVs plays a key role in upregulating Akt/survivin in DOX-treated cells.

### 2.8. MSC-sEVs Improve Cardiac LV Function from DOX-Induced Cardiomyopathy in an In Vivo Mouse Model

To explore the effects of MSC-sEVs on DOX-induced cardiomyopathy in vivo, we compared cardiac function among the control, DOX, and MSC-sEVs + DOX groups. The DOX group had significantly dilated LV chamber dimensions, and lower FS and EF than the control group, suggesting DOX-induced cardiomyopathy [32,33]. However, the MSC-sEVs + DOX group showed significantly improved LV function, as well as reduced LV chamber dimensions compared to the DOX group, similar to the control group (Figure 8, Table 1). As shown in Table 1, the MSC-sEVs + DOX group recovered both LV FS (%) and EF (%), which were decreased by DOX from 13.76 ± 0.5988 to 25.80 ± 2.191 (*p* = 0.0179) and from 35.78 ± 1.322 to 58.93 ± 3.722 (*p* = 0.0179), respectively.

### 2.9. MSC-sEVs Upregulate Both Survivin and Bcl-2 Expression in an In Vivo Mouse Model

To investigate the anti-apoptotic effects of MSC-sEVs on myocardium in the DOX-induced cardiomyopathy model, we evaluated the expression levels of survivin and Bcl-2 in the heart tissues of mice. The expression levels of both proteins decreased in the DOX group compared to the control group, but were recovered in the MSC-sEVs + DOX group (Figure 9). These results indicate that the upregulation of survivin and Bcl-2 by MSC-sEVs is involved in the anti-apoptotic effects of MSC-sEVs against DOX-induced cardiotoxicity.

## 3. Discussion

Our study confirmed that MSC-sEVs induce survivin expression in cardiomyocytes, contributing to the improvement of DOX-induced cardiotoxicity. Furthermore, we demonstrated that miR-199a-3p in MSC-sEVs regulates the Akt-Sp1/p53 pathway, which induces upregulation of survivin in DOX-treated cardiomyocytes (Figure 10).

Cellular conditions are reflected in exosomal contents such as proteins and RNAs. Similarly, MSC-sEVs including exosomes are believed to activate cellular signaling via delivering miRNAs. Furthermore, MSC-sEVs are considered a cell-free therapy for various cardiac pathological conditions following myocardial infarction, as they alleviate inflammatory reactions, accompanied by decreased infarct size [16,34]. I In addition, they may also have protective effects on DOX-induced cardiotoxicity [35]. However, the precise mechanism underlying the biological functions of MSC-sEVs in DOX-induced cardiomyopathy is not fully understood.

Co-culture of H9c2 cells with MSCs attenuates cellular senescence and cardiotoxicity [36]. Because the beneficial effects of stem cell therapy are from its paracrine effect, we speculated that the exposure of MSC-sEVs to H9c2 cells could induce a similar effect. To investigate whether MSC-sEVs have anti-apoptotic effects on DOX-induced cardiotoxicity, an in vitro experiment was performed on the MSC-sEVs + DOX treatment group. The DOX group had reduced cell viability and increased apoptosis compared to the control group, which were reversed by treatment with MSC-sEVs.

Survivin, an anti-apoptotic protein, is an important factor that inhibits cellular apoptosis in cardiomyocytes [24,37]. In addition, Akt activation by the exposure of MSC-sEVs is suggested to be a critical signal change contributing to its protective effect toward various cell types [34]. In a previous study, we showed that Akt activation, induced by an extracorporeal shock wave (ESW), induced survivin expression in cardiomyocytes [25]. Thus, we hypothesized that MSC-sEVs upregulate survivin via activation of the Akt pathway. As expected, our results showed that Akt activation induced by MSC-sEVs mediated the upregulation of survivin in cardiomyocytes. Next, cells in the MSC-sEVs + DOX group were exposed to MSC-sEVs 24 h before DOX treatment. Upregulation of survivin by pretreatment of MSC-sEVs before DOX treatment would be appropriate to examine the preventive effects of MSC-sEVs on DOX-induced cell death. Similar to the above results, additional treatment with MSC-sEVs of DOX-treated cardiomyocytes had anti-apoptotic effects, namely, cell viability and cellular apoptosis were improved and the level of Bcl-2 anti-apoptotic protein was recovered compared to the DOX-only group. Thus, pretreatment with MSC-sEVs before treatment with DOX chemotherapy may prevent cardiac toxicity.

Several transcription factors regulate the transcription of survivin in cardiomyocytes. However, it is unknown which ones are involved in regulating survivin expression after treatment with MSC-sEVs. We chose Sp1 and p53 as candidates based on a previous study that showed that these are responsible for the survivin expression induced by ESW in cardiomyocytes [25]. As expected, DOX treatment downregulated Sp1 activation and upregulated p53 activation compared to the control. However, the MSC-sEVs + DOX group showed the reverse results; the downregulated Sp1 level was recovered and the upregulated p53 level was decreased. The inhibition of Akt blocked the ability of MSC-sEVs to regulate these transcription factors with DOX treatment. Thus, Sp1 and p53 are considered key transcription factors involved in the regulation of survivin expression induced by MSC-sEVs under the DOX-treated condition in cardiomyocytes.

MSCs are considered one of the most promising cell types for the treatment of cardiovascular diseases [36]. Their therapeutic effects are attributed to their ability to regenerate or repair tissue injury. Furthermore, recent studies have suggested that their therapeutic effects are mediated by paracrine effects. Secretions from MSCs, including cytokines, chemokines, and growth factors, contribute to their potential therapeutic effects. In particular, exosomes derived from MSCs play a key role in conveying mRNAs, miRNAs, and other molecular contents to facilitate cell-to-cell communication. They also mediate the function or fate of the recipient cell. Accordingly, exosomes secreted by MSCs have the potential effect of reducing myocardial ischemia/reperfusion injury [16,34]. Similarly, we observed that MSC-sEVs had anti-apoptotic effects on DOX-induced cardiotoxicity. These results suggest that the contents of MSC-sEVs are crucial factors for inducing the cardioprotective effects of MSC-sEVs. MiRNA mediates intercellular communication via mRNA degradation or translation inhibition [17,18]. Some types of miRNAs delivered by exosomes are responsible for the change in the biological signaling pathway in recipient cells [19]. Thus, we hypothesized that some kinds of miRNAs in MSC-sEVs would modulate this anti-apoptotic effect on cardiomyocytes, the recipient cells. As MSC-sEVs reduce the apoptotic level by elevating the levels of p-Akt, survivin and Bcl-2, we selected three miRNAs (miR-199a-3p, miR-424-5p, and miR-21-5p) as candidates for major contents in MSC-sEVs contributing to their anti-apoptotic effect. Those three miRNAs are secreted from MSCs and alleviate apoptotic levels in various pathologic conditions [29,30,38,39,40]. Our data showed that overexpression of those three miRNAs elevated the level of p-Akt in cardiomyocytes under the DOX-treated condition. Similarly, overexpression of those miRNAs reversed the transcriptional activity of Sp1 and p53 induced by DOX treatment. However, regarding survivin expression, miR199a-3p is mainly involved in survivin expression in the DOX-treated condition, whereas miR424-5p and miR-21-5p have no significant effect on survivin expression under identical conditions. These results may be due to the fact that not a single miRNA affects the signaling pathway enough to induce changes; rather, groups of miRNAs induce changes.

Finally, we confirmed that consistent MSC-sEVs administration in addition to DOX improved cardiac systolic function compared to the DOX-only treated group. For the in vivo study, we chose a short-term model with a single injection of DOX (15 mg/kg, intraperitoneally) [33,41,42] and MSC-sEVs were administered every other day for the experimental period for boosting. The preliminary results showed that the effects of MSC-sEVs lasted about 24 to 48 h in cardiomyocytes. Both immunohistochemical staining and Western blotting suggested that the DOX group had a lower expression of survivin and Bcl-2 than the control group. However, the MSC-sEVs + DOX group showed significantly higher levels of those proteins compared to the DOX group. These results demonstrate the anti-apoptotic effects of MSC-sEVs on DOX-induced cardiomyopathy.

In clinical situations, DOX-induced cardiotoxicity in patients undergoing chemotherapy is one of the side effects that are very concerning, since it leads to cardiac myocyte death, progressive cardiomyopathy, and congestive heart failure. Therefore, the protective effects of MSC-sEVs against DOX-induced cardiotoxicity is expected to be of great translational value from the clinical aspect. MSC-sEVs treatment for the purpose of preventing cardiac toxicity during chemotherapy is believed to be an important therapeutic strategy in patients. However, our study has the limitation that we have not yet tested a clinically effective dose, and further studies are needed for the production of a reproducible and stable product of MSC-EVs for clinical application.

Our study demonstrated the mechanism by which MSC-sEVs protect cardiomyocytes from acute DOX-induced cardiomyopathy. The results clarify the relationship between MSC-sEVs and survivin in cardiomyocytes, and show that the miR199a-3p-Akt-Sp1/p53 pathway is involved in this mechanism. Furthermore, we suggest that the anti-apoptotic effects of MSC-sEVs improve cardiac systolic function. Our findings suggest that MSC-sEVs may be a novel cell-free therapy for patients treated with DOX to prevent cardiac toxicity.

## 4. Materials and Methods

### 4.1. Cell Culture and Reagents

The rat neonatal H9c2 cardiac myoblast cell line and murine embryonic mesenchymal stem cell line (C3H/10T1/2) were obtained from the Korea Cell Line Bank (Seoul, Korea) [43]. The H9c2 cells were cultured in Dulbecco’s modified Eagles medium (DMEM)/high glucose with 10% fetal bovine serum (FBS, HyClone, Melbourne, VIC, Australia) and 1% penicillin-streptomycin (Corning, manassas, VA, USA). The C3H/10T1/2 cells were cultured in RPMI-1640 medium with 10% FBS and 1% penicillin-streptomycin. The immortalized mouse aortic endothelial cells (iMAECs) were cultured in Dulbecco’s Modified Eagles Medium (DMEM; HyClone, South Logan, UT, USA) containing 10% FBS, 25 μg/mL endothelial cell growth supplement (BD Biosciences, Franklin Lakes, NJ, USA), and 1% penicillin-streptomycin. DOX and the phosphoinositide 3-kinase (PI3K) inhibitor LY294002 were purchased from Tocris Bioscience (Bristol, UK).

### 4.2. Isolation and Treatment of MSC-sEVs

For isolation of MSC-sEVs or iMAEC-sEVs, cells were cultured in serum-free media for 24 h. The culture medium was harvested and filtered with an 0.2 μm syringe filter (Sartorius, Sottingen, Germany) to remove apoptotic cells and microvesicles. The filtrate media were concentrated at 3000× *g* (Centrifuge 5810R; Eppendorf, Hamburg, Germany) at 4 °C for 30 min using a 100 kDa MWCO filter (Sartorius, Sottingen, Germany) for ultrafiltration-base purification of MSC-sEVs. The isolated MSC-sEVs were stored at −80 °C until use. The size and numbers of the MSC-sEVs were measured by nanoparticle tracking analysis (NTA) with the Nanosight LM10 instrument (Brunel Microscopes Ltd., Chippenham, UK). For in vitro experiments, H9c2 cells were treated with MSC-sEVs at a concentration of 3 × 10^9^.

### 4.3. Atomic Force Microscopy

The morphology of MSC-sEVs was investigated using atomic force microscopy (AFM). Briefly, one drop of MSC-sEVs suspension diluted in deionized water was absorbed on freshly cleaved AFM mica discs (Highest Grade V1; Ted Pella Inc., Redding, CA, USA) for 10 min. The sheets were thoroughly rinsed with deionized water to remove unbound MSC-sEVs and then air-dried. Micrometer-scale imaging was obtained with XE-100 AFM (Park Systems, Santa Clara, CA, USA) and processed using the PARK Systems XEI software program.

### 4.4. Animal Experiments for DOX-Induced Cardiomyopathy

Animal studies were performed according to the guidelines for animal experiments, and were approved by the Animal Experimentation Ethics Committee of Ewha Womans University (Seoul, South Korea). All animal procedures were conducted under inhalational anesthesia with 0.5 L/min oxygen and 1–2% Terrell isoflurane (Minrad International Inc., Orchard Park, NY, USA). Male C57BL/6 mice (6 weeks old; Central Animal Laboratory, Seoul, Korea) were randomly assigned to control, DOX, and MSC-sEVs + DOX groups (*n* = 8/group). As previously described, we established a model of DOX-induced cardiomyopathy in mice using the following procedures. Mice in the DOX group received a single dose of DOX (15 mg/kg, intraperitoneally [i.p.]) [33], and control mice received injections of saline of comparable volume. Mice assigned to the MSC-sEVs + DOX group received MSC-sEVs injection (4 × 10^10^, intravenously [i.v.]) at 1 day before DOX injection (15 mg/kg, i.p.), and then every other day for 14 days for boosting [44,45,46]. At 14 days after the DOX injection, echocardiographic measurements were performed and the hearts were removed from the mice.

### 4.5. Echocardiography

Echocardiography was performed using a commercially available HDI 5000 ultrasound scanner (Phillips Medical Systems, Bothell, WA, USA) with a 15 MHz linear array transducer. The animals were lightly anesthetized with 1–2% Terrell isoflurane during the echocardiographic examination. M-mode images from the parasternal long-axis view were used to measure conventional echocardiographic parameters, including the left ventricular end-diastolic dimension (LVEDD) and left ventricular end-systolic dimension (LVESD). Left ventricle (LV) function was assessed by fractional shortening (FS), and the ejection fraction (EF) was calculated from the LV linear measurements (LVEDD and LVESD).

### 4.6. Western Blotting

Cells were harvested in a lysis buffer containing 1% protease and phosphatase inhibitors. After the lysates were centrifuged at 13,000 rpm for 30 min, the supernatants were collected. The protein concentrations of the cell lysates were measured using a BCA protein assay kit (Thermo Fisher Scientific, Waltham, MA, USA). Identical amounts of proteins were subjected to sodium dodecyl sulfate-polyacrylamide gel electrophoresis and transferred to nitrocellulose membranes. The membranes were blocked in 5% skim milk in TBST for 1 h and then incubated overnight with primary antibodies. Primary antibodies were used to detect the expression of CD9 (1:1000, Abcam, Cambridge, UK), CD63 (1:1000; Santa Cruz Biotechnology, Dallas, TX, USA), CD81 (1:1000; Santa Cruz, Dallas, TX, USA), TSG101 (1:1000; Santa Cruz, Dallas, TX, USA), survivin (1:1000; Cell Signaling Technology [CST], Danvers, MA, USA), B-cell lymphoma 2 (Bcl-2) (1:500; Santa Cruz, Dallas, TX, USA), phosphorylated Akt (p-Akt) (1:1000; CST, Danvers, MA, USA), Akt (1:1000; CST), Sp1 (1:1000; EMD Millipore Corp., Temecula, CA, USA), p53 (1:1000; CST, Danvers, MA, USA), lamin A/C (1:1000; Santa Cruz, Dallas, TX, USA), and GAPDH (1:1000; Santa Cruz, Dallas, TX, USA). Total protein expression was normalized to GAPDH.

### 4.7. Cell Viability Assay and Terminal Deoxynucleotidyl Transferase dUTP Nick-End Labeling Assay

After the H9c2 cells were exposed to the indicated stimuli and incubated for 24 h, cell viability was measured using a cell viability assay kit (Abfrontier, Seoul, Korea) according to the manufacturer’s protocol. The terminal deoxynucleotidyl transferase dUTP nick-end labeling (TUNEL) assay was performed using the DeadEnd Fluorometric TUNEL System (Promega, Fitchburg, WI, USA) according to the manufacturer’s instructions. TUNEL-positive cells were visualized with the LSM 800 instrument (Carl Zeiss, Oberkochen, Germany).

### 4.8. Transfection of Small Interfering RNA

To knock down survivin expression, double-stranded target small interfering RNAs (siRNAs) were purchased from Bioneer (Daejeon, Korea). The following target sequences were used for survivin siRNA: 5′-GCA AAG GAG ACC AAC AAC AUU-3′ and 5′-UGU UGU UGG UCU CCU UUG CUU-3.’ Transfection of siRNA was performed using Lipofectamine RNAiMAX (Invitrogen, Carlsbad, CA, USA) according to the manufacturer’s protocols. At 24 h after siRNA transfection, the H9c2 cells were exposed to MSC-sEVs followed by treatment with 1 μM DOX for 24 h.

### 4.9. Nuclear/Cytosol Fractionation

Nuclear and cytosolic fractions were prepared using a nuclear/cytosol fractionation kit (Biovision, Milpitas, CA, USA) according to the manufacturer’s instructions. The purity of the protein fractions was assessed by immunoblotting the fractions with anti-lamin A/C (nuclear protein) and anti-GAPDH (cytoplasmic protein) antibodies.

### 4.10. Validation of miRNAs by Quantitative PCR

To confirm the differential expression of miRNAs between iMAEC-sEVs and MSC-sEVs, we performed quantitative PCR (qPCR) for miRNAs that were highly differentially expressed between iMAEC-sEVs and MSC-sEVs (3 miRNAs from previous reports). Total RNA from iMAEC-sEVs and MSC-sEVs was purified using the miRNeasy mini kit (Qiagen, Hilden, Germany) according to the manufacturer’s instructions. The RNA concentration was quantified using a NanoDrop (ND-2000) spectrophotometer (Thermo Fisher Scientific). Complementary DNA was synthesized from 400 ng total RNA using the miScript II RT Kit. Quantification of miRNAs was performed using the miScript SYBR Green PCR Kit (Qiagen) on the ABI StepOne Real-Time PCR system according to the manufacturer’s instructions (Applied Biosystems, Foster City, CA, USA). Furthermore, three independent miRNA levels were measured to identify their differential levels in MSC-sEVs and to confirm their overexpression in H9c2 cells. In these experiments, synthetic *Caenorhabditis elegans* miRNA cel-miR-39 (Qiagen, Hilden, Germany) was spiked into each sample as an internal control for RNA isolation. All reactions were performed in triplicate. The concentrations of miRNAs in the samples were calculated based on their Ct values normalized to cel-miR-39.

### 4.11. Transfection of miRNA Mimics and miRNA Inhibitors in Cardiomyocytes

For overexpression or inhibition of miRNAs, miRNA mimics and miRNA inhibitors targeting miR-199a-3p, miR-424-5p, and miR-21-5p were purchased from Genolution Pharmaceuticals (Seoul, Korea). The following target gene information was used: mmu-miR-199a-3p (MIMAT0000230), mmu-miR-424-5p (MIMAT0000548), and mmu-miR-21-5p (MIMAT0000530). After 2 h of starvation in serum-free media, transfection of miRNA mimics and miRNA inhibitors was performed using Lipofectamine RNAiMAX (Invitrogen) in H9c2 cells according to the manufacturer’s protocols.

### 4.12. Immunofluorescence Staining

The heart tissues of the mice were collected for immunofluorescence staining 14 days after DOX injection. Then, the heart tissues, specifically the myocardial regions, were sectioned at 5 μm thickness. All animal procedures were approved by the Research Ethics Committees of Ewha Womans University and conducted in accordance with approved guidelines. After fixation in 4% paraformaldehyde (Biosesang Inc., Seongnam, Korea), the tissues were blocked in 10% normal goat serum and incubated overnight at 4 °C with anti-mouse Bcl-2 antibody (1:100; Santa Cruz) and anti-rabbit survivin antibody (1:400; CST, Danvers, MA, USA). Anti-mouse AlexaFluor 647 goat (Life Technologies, Carlsbad, CA, USA) and anti-rabbit AlexaFluor 568 goat antibodies (Life Technologies, Carlsbad, CA, USA) were used as the secondary antibodies. After nuclear counter-staining with DAPI, the tissue slides were mounted and visualized with the LSM 800 confocal microscope.

### 4.13. Statistical Analysis

All data are expressed as the mean ± standard error of the mean (SEM) of at least three independent experiments. The nonparametric Mann–Whitney U-test was used for testing differences in quantitative variables. Those *p* values less than 0.05 were considered statistically significant.

## 5. Conclusions

Our results suggest that MSC-sEVs have potentially protective effects against acute DOX-induced cardiomyopathy, through mechanisms that upregulate surviving expression via regulating Akt-Sp1/p53 signaling pathway. Our findings suggest that MSC-sEVs may be a novel cell-free therapy for patients treated with DOX to prevent cardiac toxicity.

## Figures and Tables

**Figure 1 ijms-22-07102-f001:**
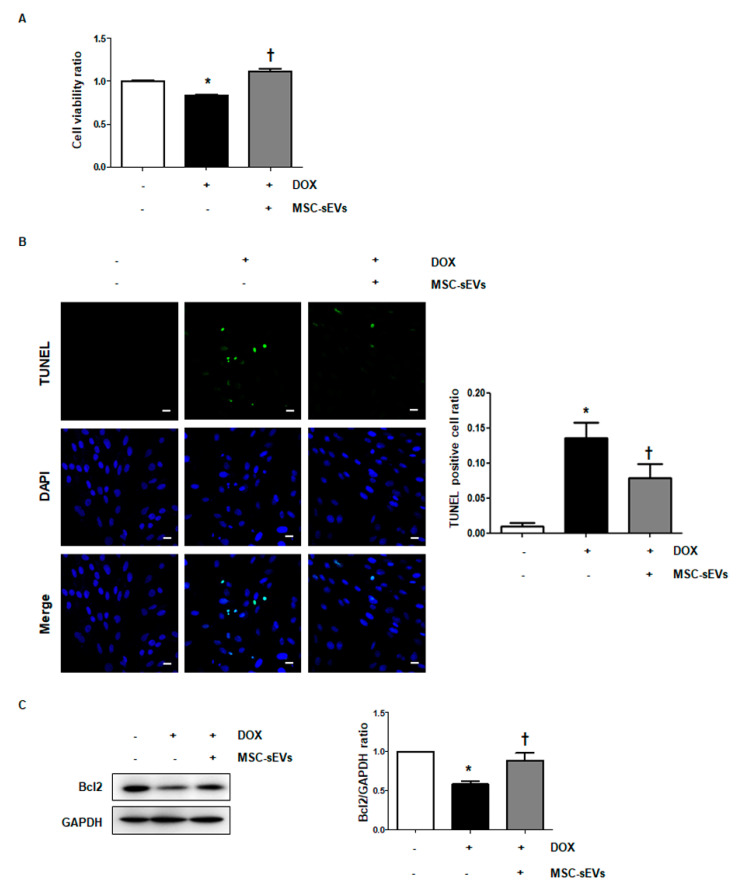
MSC-sEVs protect cardiomyocytes from DOX−induced cell death. H9c2 cells were pretreated with MSC-sEVs (3 × 10^9^) for 24 h prior to DOX (1 μM) for 24 h. (**A**) Cell viability was measured using the MTT assay (*n* = 5). * Significant difference compared to the control (*p* < 0.05). † Significant difference compared to the DOX treatment (*p* < 0.05). (**B**) The bar graph shows the TUNEL−positive cell ratio as a cellular apoptotic index. TUNEL−stained cells (green) were counted and normalized to DAPI−stained cells (blue): control (0.0097 ± 0.0048, mean ± SEM), DOX (0.1361 ± 0.0220, mean ± SEM) and MSC-sEVs +DOX (0.0784 ± 0.0207, mean ± SEM) (*n* = 5). * Significant difference compared to the control (*p* < 0.05). † Significant difference compared to DOX treatment (*p* < 0.05). Representative fluorescence microscopic TUNEL images from at least three independent experiments are shown (magnification, 200×; scale bars, 10 μm). (**C**) Protein levels of Bcl−2 were measured by Western blotting (*n* = 5). * Significant difference compared to the control (*p* < 0.05). † Significant difference compared to DOX treatment (*p* < 0.05).

**Figure 2 ijms-22-07102-f002:**
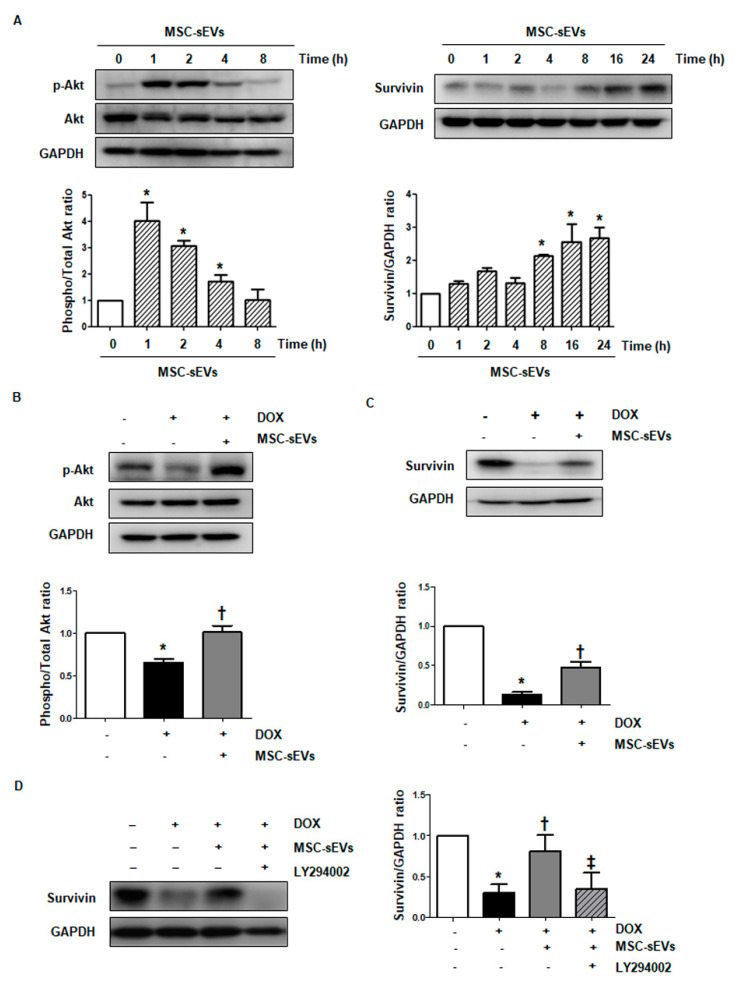
MSC-sEVs inhibit DOX−induced downregulation of survivin via the activation of Akt in cardiomyocytes. (**A**) H9c2 cells were treated with MSC-sEVs (3 × 10^9^) for the indicated times. The protein levels of p−Akt, Akt, survivin, and GAPDH were measured by Western blotting. * Significant difference compared to the control (*p* < 0.05). (**B**,**C**) H9c2 cells were cultured in DOX (1 μM) for 8 or 24 h after 24 h of exposure to MSC-sEVs. The protein levels of *p*−Akt, Akt, survivin, and GAPDH were measured by Western blotting (*n* = 5). * Significant difference compared to the control (*p* < 0.05). † Significant difference compared to DOX treatment (*p* < 0.05). (**D**) After treatment with LY294002 (50 μM) for 2 h, the cells were exposed to MSC-sEVs (3 × 10^9^) for 24 h and then additionally treated with DOX (1 μM) for 24 h. The protein levels of survivin and GAPDH were measured by Western blotting (*n* = 5). * Significant difference compared to the control (*p* < 0.05). † Significant difference compared to DOX treatment (*p* < 0.05). ‡ Significant difference compared to MSC-sEVs + DOX treatment (*p* < 0.05).

**Figure 3 ijms-22-07102-f003:**
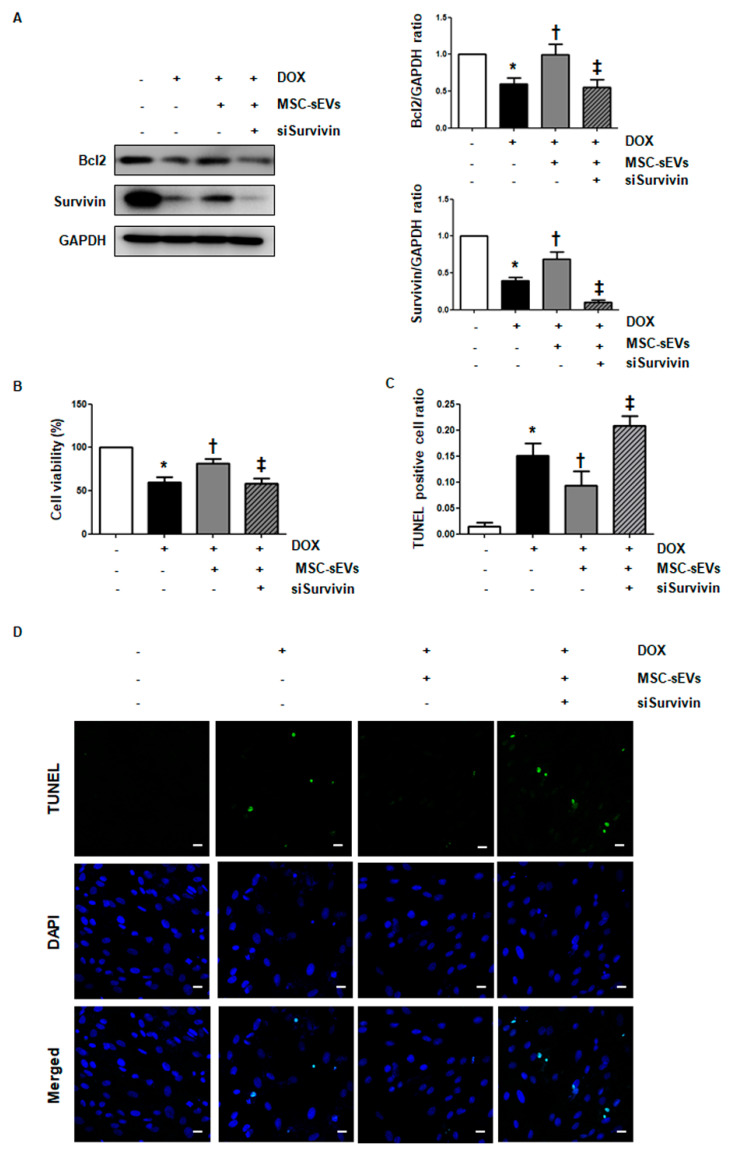
Knockdown of survivin alleviates the cardioprotective effects of MSC-sEVs on DOX−induced cell death. H9c2 cells transfected with siRNA (50 nM) targeting survivin were exposed to MSC-sEVs (3 × 10^9^) for 24 h and then treated with DOX for 24 h. (**A**) The protein levels of survivin, Bcl−2, and GAPDH were measured by Western blotting (*n* = 5). * Significant difference compared to the control (*p* < 0.05). † Significant difference compared to DOX treatment (*p* < 0.05). ‡ Significant difference compared to MSC-sEVs + DOX treatment (*p* < 0.05). (**B**) Cell viability of each condition was measured using the MTT assay. * Significant difference compared to the control (*p* < 0.05). † Significant difference compared to DOX treatment (*p* < 0.05) ‡ Significant difference compared to MSC-sEVs + DOX treatment (*p* < 0.05). (**C**,**D**) Cellular apoptosis was measured using the TUNEL assay. Representative fluorescence TUNEL images from at least three independent experiments are shown (magnification, 200×; scale bars, 10 μm). The bar graph shows the TUNEL−positive cell ratio as a cellular apoptotic index. TUNEL−stained cells (green) were counted and normalized to DAPI−stained cells (blue) (*n* = 5). * Significant difference compared to the control (*p* < 0.05). † Significant difference compared to DOX treatment (*p* < 0.05). ‡ Significant difference compared to MSC-sEVs + DOX treatment (*p* < 0.05).

**Figure 4 ijms-22-07102-f004:**
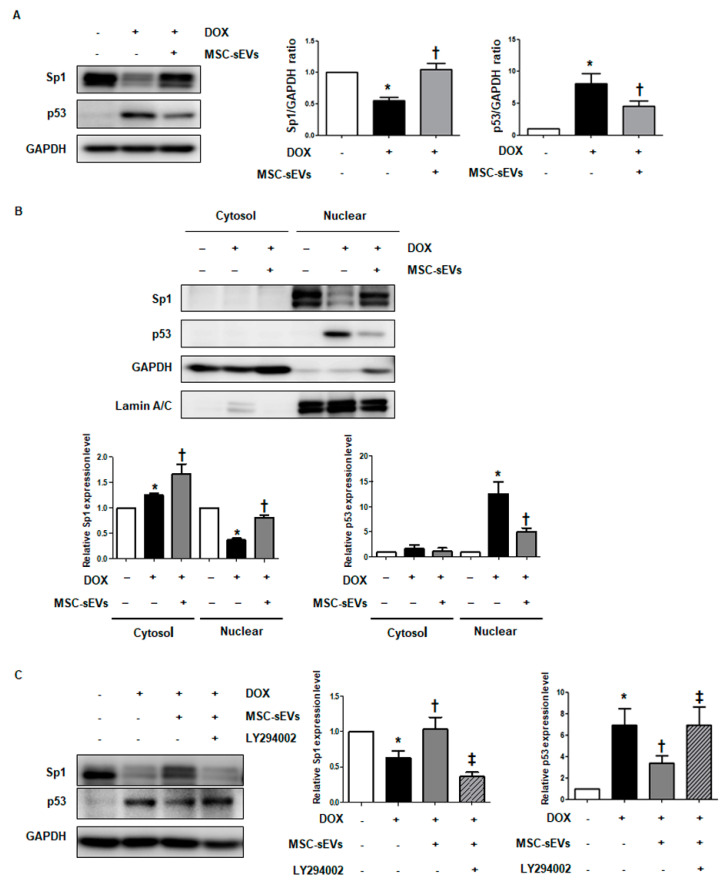
Akt activation by MSC-sEVs induces survivin by downregulating p53 and upregulating Sp1. H9c2 cells were pretreated with MSC-sEVs for 24 h and then incubated with DOX (1 μM) for 24 h. (**A**) The protein levels of Sp1, p53, and GAPDH were measured by Western blotting (*n* = 5). * Significant difference compared to the control (*p* < 0.05). † Significant difference compared to DOX treatment (*p* < 0.05). (**B**) Cytoplasmic and nuclear extracts were isolated from H9c2 cells under each condition. The protein levels of Sp1, p53, GAPDH, and lamin A/C were detected by Western blotting (*n* = 5). GAPDH and lamin A/C were used as internal controls for the cytosolic and nuclear fractions, respectively. * Significant difference compared to the control (*p* < 0.05). † Significant difference compared to DOX treatment (*p* < 0.05). (**C**) After pretreatment with LY294002 (50 μM) for 2 h, H9c2 cells were exposed to MSC-sEVs for 24 h and then treated with DOX (1 μM) for 24 h. The protein levels of Sp1, p53, and GAPDH were measured by Western blotting (*n* = 5). * Significant difference compared to the control (*p* < 0.05). † Significant difference compared to DOX treatment (*p* < 0.05). ‡ Significant difference compared to MSC-sEVs + DOX treatment (*p* < 0.05).

**Figure 5 ijms-22-07102-f005:**
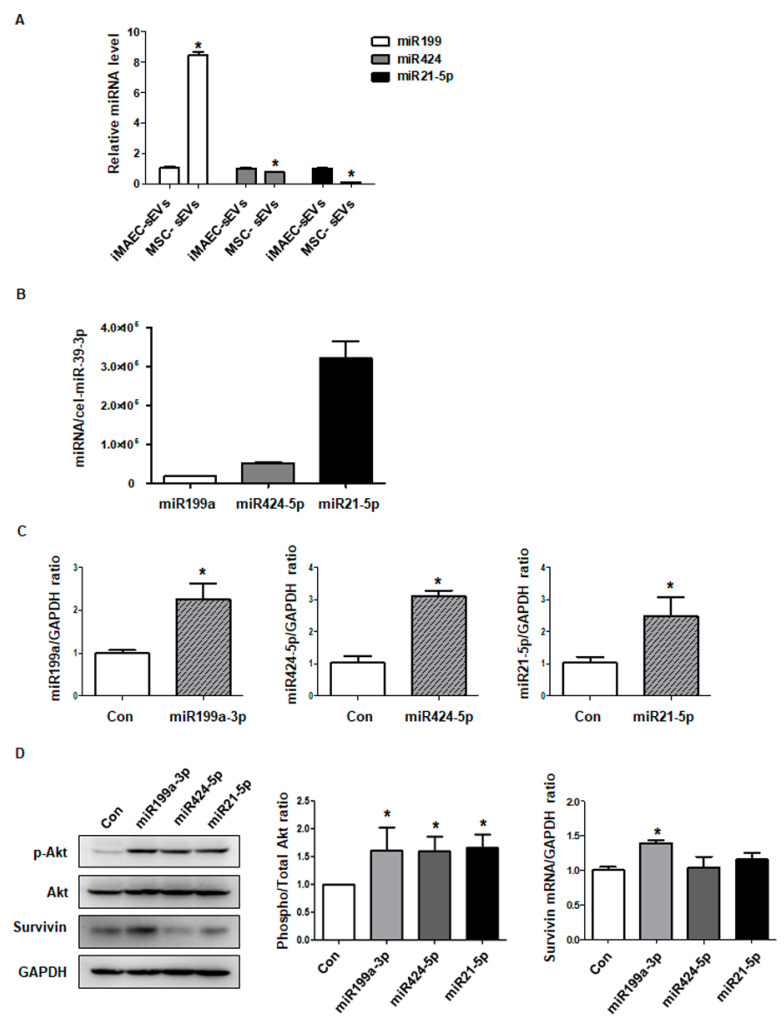
MiRNAs in MSC-sEVs contribute to the Akt/survivin pathway in cardiomyocytes. (**A**) The levels of miR199a−3p, miR424−5p and miR21−5p in both iMAEC−sEVs and MSC-sEVs were measured with qPCR. The miRNA levels of MSC-sEVs are indicated relative to those of the iMAEC−sEVs (*n* = 5). * Significant difference compared to the iMAEC−sEVss (*p* < 0.05). (**B**) The levels of three miRNAs (miR199a−3p, miR424−5p, and miR 21−5p) in MSC-sEVs were measured by qPCR (*n* = 5). Synthetic *Caenorhabditis elegans* miRNA, cel−miR−39 was spiked into each sample as an internal control. (**C**) For overexpression of three miRNAs (miR199a−3p, miR424−5p, and miR 21−5p), H9c2 cells were transfected with three individual miRNA mimics. The levels of three miRNAs were measured with qPCR (*n* = 5). * Significant difference compared to the control (*p* < 0.05). (**D**) H9c2 cells transfected with three independent miRNAs (miR199a−3p, miR424−5p, and miR 21−5p) were harvested. The protein levels of p−Akt, Akt, survivin, and GAPDH were measured by Western blotting (*n* = 5). * Significant difference compared to the control (*p* < 0.05).

**Figure 6 ijms-22-07102-f006:**
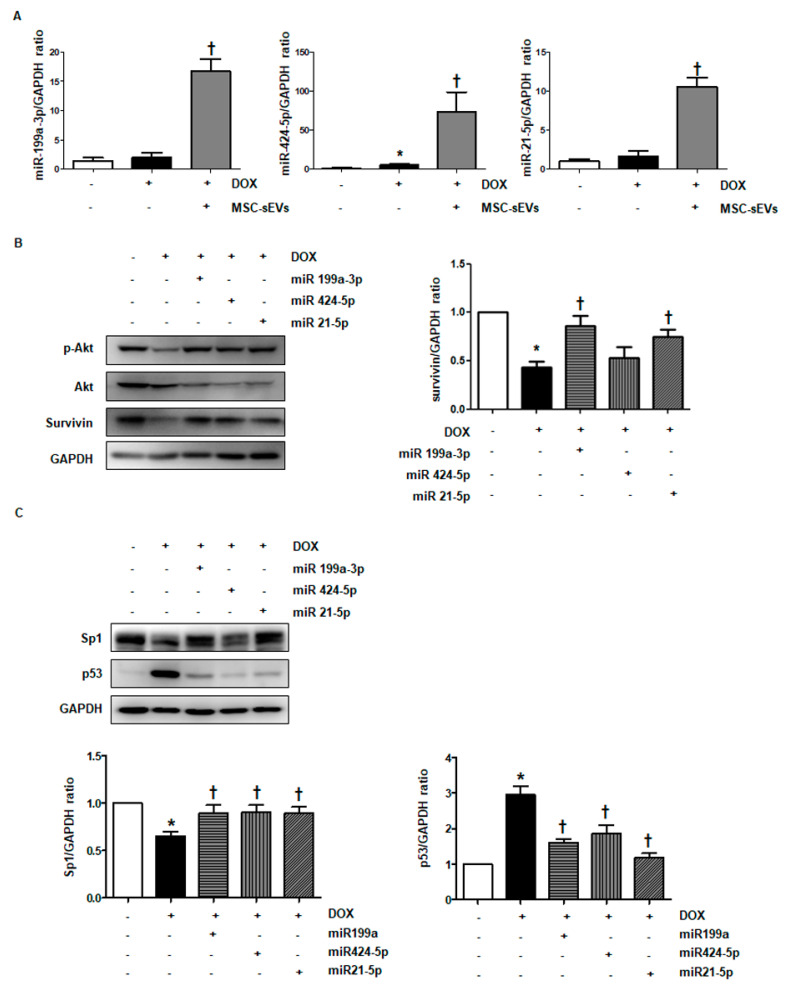
MiR−199a−3p and miR−21−5p in MSC-sEVs are involved in regulating the Akt−Sp1/p53−survivin axis in DOX−treated cardiomyocytes. (**A**) H9c2 cells were pretreated with MSC-sEVs (3 × 10^9^) for 24 h and then incubated with DOX (1 μM) for 24 h. The levels of miR199a−3p, miR424−5p and miR21−5p were measured by qPCR (*n* = 5). * Significant difference compared to the control (*p* < 0.05). † Significant difference compared to DOX treatment (*p* < 0.05). (**B**,**C**) H9c2 cells transfected with three miRNA mimics were incubated with DOX for 24 h. The protein levels of p−Akt, Akt, survivin, Sp1, p53 and GAPDH were measured by Western blotting (*n* = 5). * Significant difference compared to the control (*p* < 0.05). † Significant difference compared to DOX treatment (*p* < 0.05).

**Figure 7 ijms-22-07102-f007:**
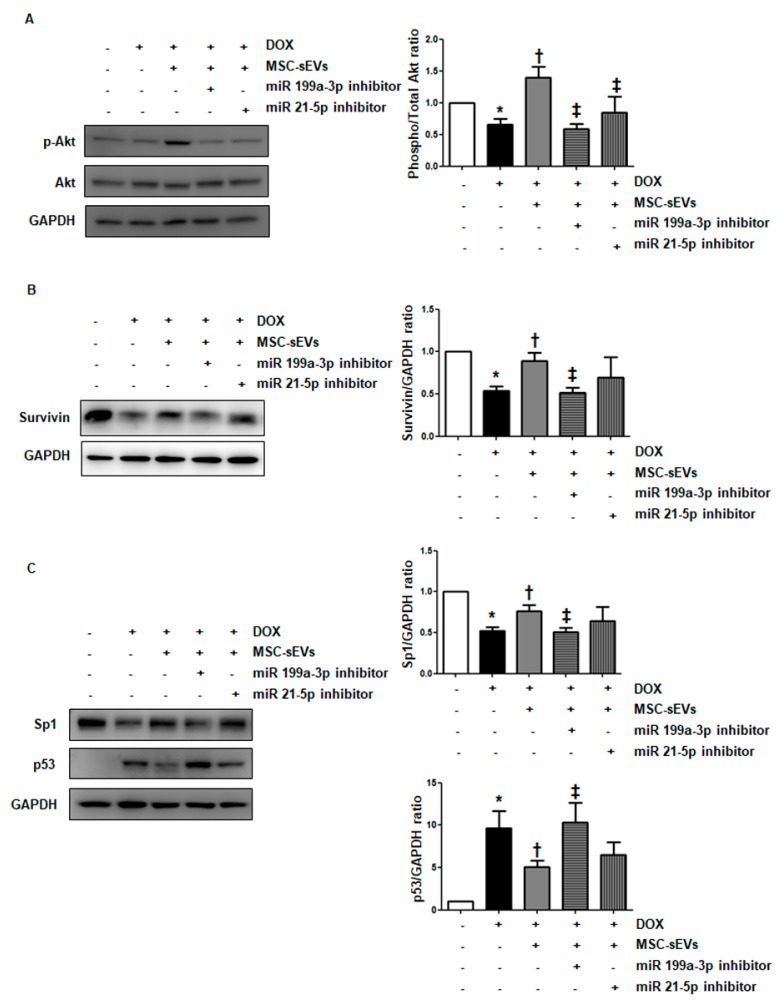
Inhibition of miR−199a−3p alleviates the effects of MSC-sEVs on the Akt−Sp1/p53−survivin axis in DOX−treated cardiomyocytes. For inhibition of miR−199a−3p and miR−21−5p, H9c2 cells were transfected with miRNA inhibitors (100 nM) targeting each miRNA. (**A**) After treatment with MSC-sEVs (3 × 10^9^) for 1 h, cells were treated with DOX (1 μM) for 8 h. The protein levels of p−Akt, Akt, and GAPDH were measured by Western blotting (*n* = 5). (**B**,**C**) H9c2 cells transfected with miRNA inhibitors were pre−treated with MSC-sEVs (3 × 10^9^) for 24 h, and then incubated with DOX (1 μM) for another 24 h. The protein levels of survivin, Sp1, p53, and GAPDH were measured by Western blotting (*n* = 5). * Significant difference compared to the control (*p* < 0.05). † Significant difference compared to DOX treatment (*p* < 0.05). ‡ Significant difference compared to MSC-sEVs + DOX treatment (*p* < 0.05).

**Figure 8 ijms-22-07102-f008:**
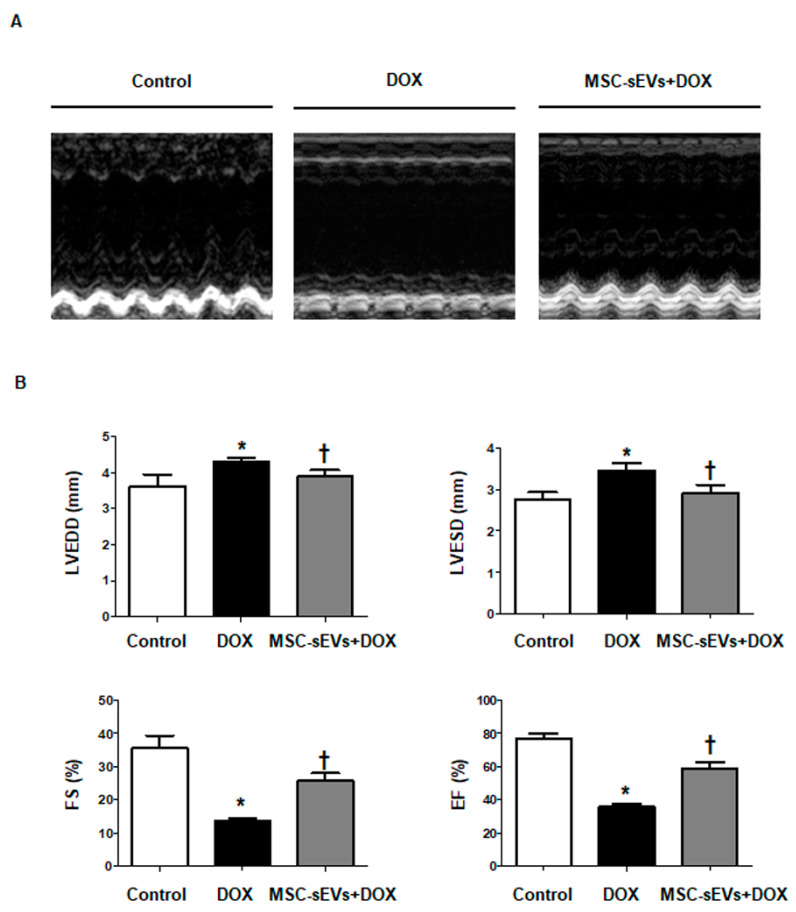
MSC-sEVs improves acute DOX−induced cardiomyopathy in an in vivo mouse model. (**A**) Representative M−mode images from the control, DOX, and MSC-sEVs + DOX groups. (**B**) Quantitative echocardiographic group data: LVEDD (mm), LVESD (mm), FS (%) and EF (%) (*n* = 8/group). * Significant difference compared to the control (*p* < 0.05). † Significant difference compared to DOX treatment (*p* < 0.05).

**Figure 9 ijms-22-07102-f009:**
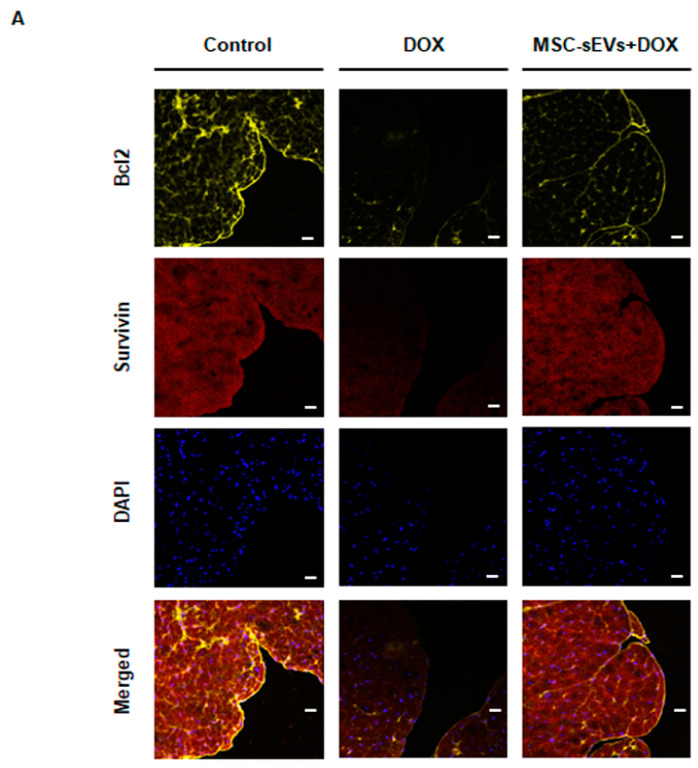
MSC-sEVs attenuate cardiomyocyte apoptosis by upregulation of anti−apoptotic proteins (survivin and Bcl2) in an in vivo mouse model. (**A**) Immunofluorescence staining to detect Bcl−2 and survivin in various regions of the mouse myocardium. The heart tissues were stained with anti−Bcl−2 antibody (yellow) and anti−survivin antibody (red), while nuclei were stained with DAPI (blue). The expression levels of Bcl−2 and survivin were compared among the three groups (control, DOX and MSC-sEVs + DOX) using confocal microscopy. Representative images are shown (magnification 400×; scale bars, 20 μm). (**B**) The protein levels of survivin, Bcl−2, and GAPDH in the heart tissue of mice were measured by Western blotting (*n* = 8/group). Representative images are shown. * Significant difference compared to the control (*p* < 0.05). † Significant difference compared to DOX treatment (*p* < 0.05).

**Figure 10 ijms-22-07102-f010:**
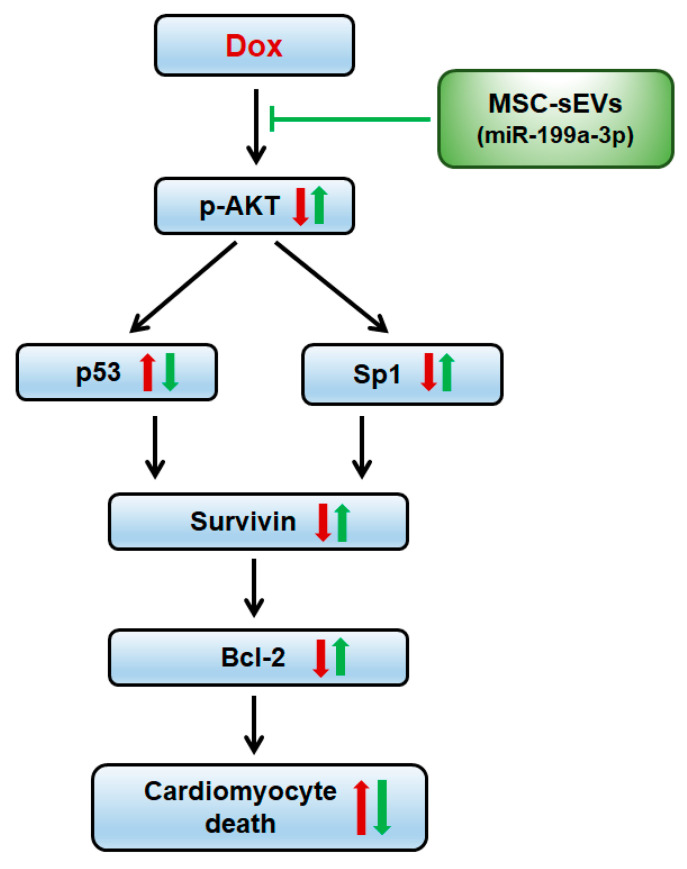
Schematic of the proposed role of MSC-sEVs in DOX−induced cardiomyopathy. In cardiomyocytes, DOX induces an increase in the transcription factor p53 and a decrease in Sp1 through Akt inhibition, thereby inducing cardiomyocyte death by downregulating the expression of survivin and Bcl2 related to cell survival. The miR−199a−3p in MSC-sEVs plays a key role in the protective effects of MSC-sEVs against DOX−induced cardiomyopathy, by upregulating survivin expression via regulating the Akt−Sp1/p53 signaling pathway.

**Table 1 ijms-22-07102-t001:** MSC-sEVs improve both dilated cardiomyopathy and cardiac functions induced by DOX in an in vivo mouse model. Echocardiography was performed in the control, DOX, and MSC-sEVs + DOX groups 14 days after DOX injection. Quantitative echocardiographic group data: LVEDD (mm), LVESD (mm), FS (%) and EF (%) (*n* = 8/group). Values are mean ± SEM. Each *p*-value is the result of a comparison between two groups (control vs. DOX, DOX vs. MSC-sEVs + DOX).

Variable	Control	DOX	MSC-sEVs + DOX	*p* Value (Control vs. DOX)	*p* Value (DOX vs. MSC-sEVs + DOX)
LVEDD (mm)	3.625 ± 0.3326	4.317 ± 0.1014	3.900 ± 0.1732	0.0268	0.0460
LVESD (mm)	2.767 ± 0.1667	3.463 ± 0.1580	2.900 ± 0.2082	0.0319	0.0407
FS (%)	35.54 ± 3.738	13.76 ± 0.5988	25.80 ± 2.191	0.0079	0.0179
EF (%)	76.63 ± 3.161	35.78 ± 1.322	58.93 ± 3.722	0.0179	0.0179

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
