# Peer review of "Mesenchymal Stem Cell-Derived Small Extracellular Vesicles Protect Cardiomyocytes from Doxorubicin-Induced Cardiomyopathy by Upregulating Survivin Expression via the miR-199a-3p-Akt-Sp1/p53 Signaling Pathway"

_ijms, 2021, doi:10.3390/ijms22137102_

Round 1

Reviewer 1 Report

In this interesting work, authors showed, on the animal model, that administration of small extracellular vesicles derived from mesenchymal stem cells was able to ameliorate cardiac dysfunction induced by doxorubicin in the experimental mice model.

The work has been carried out properly, methods are clearly stated and results are well-substantiated. I have only minor comments:

  1. I would strongly advise authors to create and add a summarizing figure to their work showing the mechanistic rationale and molecular signaling pathway in which survivin helps to prevent apoptosis of cardiomyocytes and how this treatment improved cardiac function.
  2. Authors should shed some light on the clinical relevance of their results and their potential translational value. This should be briefly discussed in the Discussion. Was similar work done? Was this type of treatment evaluated in any of the clinical studies?
  3.  

Author Response

# Response to Reviewer 1 Comments

In this interesting work, authors showed, on the animal model, that administration of small extracellular vesicles derived from mesenchymal stem cells was able to ameliorate cardiac dysfunction induced by doxorubicin in the experimental mice model.

The work has been carried out properly, methods are clearly stated and results are well-substantiated. I have only minor comments:

Point 1. I would strongly advise authors to create and add a summarizing figure to their work showing the mechanistic rationale and molecular signaling pathway in which survivin helps to prevent apoptosis of cardiomyocytes and how this treatment improved cardiac function.

Response 1: We appreciate the constructive comments. As suggested, we added a summarizing figure in Figure 10 of manuscript (See p18) as following.

Figure 10. Schematic of the proposed role of MSC-sEVs in Dox-induced cardiomyopathy. In cardiomyocytes, Dox induces an increase in the transcription factor p53 and a decrease in Sp1 through Akt inhibition, thereby inducing cardiomyocyte death by downregulating the expression of survivin and Bcl2 related to cell survival. The miR-199a-3p in MSC-sEVs plays key role in protective effects of MSC-sEVs against Dox-induced cardiomyopathy by upregulating survivin expression via regulating Akt-Sp1/p53 signaling pathway.

Point 2. Authors should shed some light on the clinical relevance of their results and their potential translational value. This should be briefly discussed in the Discussion. Was similar work done? Was this type of treatment evaluated in any of the clinical studies?

Response 2: We appreciate the constructive comments. We descripted the translational value of MSC-sEVs treatment in patients receiving chemotherapy in discussion (See p19-p20) as following.

“In clinical, Dox-induced cardiotoxicity in patients undergoing chemotherapy is one of the side effects that are very concerned, leads to cardiac myocyte death, progressive cardiomyopathy, and congestive heart failure. Therefore, the protective effects of MSC-sEVs against Dox-induced cardiotoxicity is expected to be a great translational value in aspect of clinical. MSC-sEVs treatment for the purpose of preventing cardiac toxicity during chemotherapy is believed to be an important therapeutic strategy in patients.”

We searched for data on clinical trials on cardiotoxicity using MSC-exosomes, but no clinical trials have been conducted yet. Here, we attached review papers of the most recent clinical trials of MSC-exosomes.

(Journal of Clinical Medicine 2021;10:711, https://doi.org/10.3390/jcm10040711)

(Journal of Translational Medicine 2020;18:449, https://doi.org/10.1186/s40364-019-0159-x)

Reviewer 2 Report

In this article, authors proposed the use of Mesenchymal Stem Cell-derived exosomes to protect cardiomyocytes from doxorubicin-induced cardiomyopathy.

Presented results are very interesting. However, in my opinion, the manuscript needs some revisions.

Comments:

  • Authors isolated iMAEC-sEVs. Is not clear if this is another name of MSC-sEVs or another type of vesicles. The complete name was not written.
  • Why do you used only the exosomes and not the microvescicles or all secretome?
  • Do you evaluate the mean loss of vesicles during the freezing step?
  • Overall, the presented isolation method was not scalable. There are many paper that showed the importance to use scalable isolation method (for example for clinical applications). The use of all secretome could reduce the manipulation of the product. I suggest to evaluate this possibility for future studies.
  • I reported some studies that explain these aspects:

Agrahari et al. (2018) Trends in Biotechnol Doi: 10.1016/j.tibtech.2018.11.012

Reiner et al. (2017) Stem Cells Translational Medicine 6:1730-1739

Bari et al. (2018) Cells 7, 190; doi:10.3390/cells7110190

Gardiner et al. (2016) Journal of Extracellular Vesicles 5: 32945

Bari et al. (2020). European Journal of Pharmaceutics and Biopharmaceutics 155: 37-48

  • Conclusion section is equal to final part of discussion. Please modified.
  • The feasibility to produce a reproducible and stable product was a critical point for the clinical use of MSC-EVs. Do you evaluate the effective dose? Please add these aspects in the manuscript.

Author Response

# Response to Reviewer 2 Comments

Point 1: Authors isolated iMAEC-sEVs. Is not clear if this is another name of MSC-sEVs or another type of vesicles. The complete name was not written.

Response 1: We appreciate your comments. We added the complete name of iMAECs-sEVs in result (See p10, Section of 2.6) and Material and Methods (See p20, Section of 4.1) as following.

“To identify whether those three miRNAs are indeed representative of MSC-sEVs compared to sEVs from other types of cells, we compared their levels between immortalized mouse aortic endothelial cells-derived small extracellular vesicles (iMAEC-sEVs) and MSC-sEVs.”

 “The immortalized mouse aortic endothelial cells (iMAECs) were cultured in Dulbecco’s Modified Eagles Medium (DMEM; HyClone) containing 10% FBS, 25 ug/mL endothelial cell growth supplement (BD Biosciences, Franklin Lakes, NJ, USA), and 1% penicillin-streptomycin.

Point 2: Why do you used only the exosomes and not the microvescicles or all secretome?

Response 2: Mesenchymal stem cell (MSC) are being exploited as an experimental therapy for a variety of human diseases because they improve diseases by inhibiting inflammation and promoting regeneration. Recent studies, MSCs also produce extracellular vesicles of varying sizes including exosomes that carry as cargo mRNAs, microRNAs, and proteins. In this study, we focused on effects and functional mechanisms of MSC-exosomes because numerous studies have reported that MSC-exosome function as paracrine mediators in tissue repair and recapitulate to a large extent the therapeutic effects of parental MSCs.

Point 3: Do you evaluate the mean loss of vesicles during the freezing step?

Response 3: This is a critical point in storage of exosome. According to recently reported study on exosome storage, the different storage temperature and period influences recovery yield and morphology of exosome. The storage at below -70°C that had no significant effect either on EV number or size is the favorable condition for preservation of fresh exosomes for clinical application and basic researches (Journal of Extracellular Vesicles 2014;3:10 .3402/jev.v3.25465, Biotechnology and Bioprocess Engineering 2016;21:299-304).

We also tested using Nanosight Tracking Analysis (NTA), the number of exosomes before and after freezing was the same.

Point 4: Overall, the presented isolation method was not scalable. There are many paper that showed the importance to use scalable isolation method (for example for clinical applications). The use of all secretome could reduce the manipulation of the product. I suggest to evaluate this possibility for future studies.

I reported some studies that explain these aspects:

Agrahari et al. (2018) Trends in Biotechnol Doi: 10.1016/j.tibtech.2018.11.012

Reiner et al. (2017) Stem Cells Translational Medicine 6:1730-1739

Bari et al. (2018) Cells 7, 190; doi:10.3390/cells7110190

Gardiner et al. (2016) Journal of Extracellular Vesicles 5: 32945

Bari et al. (2020). European Journal of Pharmaceutics and Biopharmaceutics 155: 37-48

Response 4: We appreciate the constructive comments. As your comment, our isolation method is a small scale to prove the roles and action mechanism of MSC-EVs in chemotherapy-induced cardiac toxicity. For clinical application, there are still many challenges to be addressed concerning the scalable production, standardization, and characterization of MSC-EV products for the successful translation of EV-based therapeutics in the clinic. Therefore, we are also planning to set up a cell bioreactor and separation filter system for scalable production of EVs, and your suggestion is considered to be a very important part for further studies.

Point 5: Conclusion section is equal to final part of discussion. Please modified.

Response 5: We appreciate the constructive comments. As suggested, we modified conclusion section as following (See p22, Section of 5) as following. 

 Our results suggest that MSC-sEVs has potentially protective effects against acute DOX-induced cardiomyopathy through mechanisms that upregulates surviving expression via regulating Akt-Sp1/p53 signaling pathway. Our findings suggest that MSC-sEVs may be a novel cell-free therapy for patients treated with DOX to prevent cardiac toxicity.”

Point 6: The feasibility to produce a reproducible and stable product was a critical point for the clinical use of MSC-EVs. Do you evaluate the effective dose? Please add these aspects in the manuscript.

Response 6: We appreciate the constructive comments. Our study is still experimentally to prove the roles and action mechanism of MSC-EVs in chemotherapy-induced cardiac toxicity, so we have not yet tested a clinically effective dose.

As your comment, the feasibility to produce a reproducible and stable product was a critical point for the clinical use of MSC-EVs. In these parts, there are issued as quality controls including the scalable production, standardization, and characterization of MSC-EVs products for the successful translation of EVs-based therapeutics. Therefore, it will be needed to further study for production of a reproducible and stable product of MSC-EVs. These aspects were described in discussion as following (See p20) as following.

However, our study have limitation that we have not yet tested a clinically effective dose. And further study are needed for production of a reproducible and stable product of MSC-EVs for clinical application.”

Round 2

Reviewer 2 Report

Authors modified the manuscript as requested.